# Exploring the Gut Microbiota and Cardiovascular Disease

**DOI:** 10.3390/metabo11080493

**Published:** 2021-07-29

**Authors:** Kiera Murphy, Aoife N. O’Donovan, Noel M. Caplice, R. Paul Ross, Catherine Stanton

**Affiliations:** 1Teagasc Food Research Centre, Moorepark, Co. Cork P61 C996, Ireland; kiera.healy@teagasc.ie (K.M.); aoife.odonovan@teagasc.ie (A.N.O.); 2APC Microbiome Ireland, Biosciences Institute, University College Cork, Cork T12 YT20, Ireland; n.caplice@ucc.ie (N.M.C.); p.ross@ucc.ie (R.P.R.); 3VistaMilk SFI Research Centre, Teagasc, Moorepark, Co. Cork P61 C996, Ireland; 4Centre for Research in Vascular Biology, Biosciences Institute, University College Cork, Cork T12 YT20, Ireland

**Keywords:** cardiovascular disease, gut microbiota, metabolites, probiotics, faecal microbiota transplantation

## Abstract

Cardiovascular disease (CVD) has been classified as one of the leading causes of morbidity and mortality worldwide. CVD risk factors include smoking, hypertension, dyslipidaemia, obesity, inflammation and diabetes. The gut microbiota can influence human health through multiple interactions and community changes are associated with the development and progression of numerous disease states, including CVD. The gut microbiota are involved in the production of several metabolites, such as short-chain fatty acids (SCFAs), bile acids and trimethylamine-N-oxide (TMAO). These products of microbial metabolism are important modulatory factors and have been associated with an increased risk of CVD. Due to its association with CVD development, the gut microbiota has emerged as a target for therapeutic approaches. In this review, we summarise the current knowledge on the role of the gut microbiome in CVD development, and associated microbial communities, functions, and metabolic profiles. We also discuss CVD therapeutic interventions that target the gut microbiota such as probiotics and faecal microbiota transplantation.

## 1. Introduction

Cardiovascular disease (CVD) is a general term for a number of pathologies including coronary heart disease (CHD), cerebrovascular disease, peripheral artery disease, congenital and rheumatic heart disease and venous thromboembolism [1]. CVD is a chronic progressive condition, often leading to irreversible damage to vascular structures in the form of atherosclerosis and detrimental clinical outcomes such as arterial thrombosis, myocardial infarction (MI) and stroke. Elevated levels of serum cholesterol (hypercholesterolemia), particularly low-density lipoprotein cholesterol (LDL-C) is a well-documented risk factor for CVD. LDL-C deposits cholesterol into artery walls and this plaque build-up can lead to atherosclerosis [2]. CVD has been classified as one of the leading causes of morbidity and mortality worldwide, which has led to extensive health and economic burdens accounting for approximately one in every three deaths in the United States and one in every four deaths in Europe [3].

The World Health Organisation (WHO) has estimated that over 75% of premature CVD is avoidable and that risk factor improvement can aid in the reduction in the growing incidence of CVD [4]. The INTERHEART study established a case–control study of acute MI from 52 countries worldwide. The study identified nine easily assessed risk factors which were highly and significantly associated with an increased risk of acute MI [5]. Smoking, dyslipidaemia, hypertension, diabetes, abdominal obesity, psychosocial factors, daily fruits and vegetables, exercise, alcohol intake and apolipoprotein (Apo)B/ApoA1 ratio accounted for over 90% of the risk of acute MI.

The human body harbours trillions of microbial cells that form a complex microbial community in the intestine known as the gut microbiota. The gut is predominantly inhabited by two phyla, Bacteriodetes and Firmicutes, which account for 90% of the total inhabiting microbes. The remaining 10% consists of Actinobacteria, Cyanobacteria, Fusobacteria, Proteobacteria and Verrucomicrobia [6,7]. Beginning at birth, there are multiple factors that can influence how the gut is colonised, such as mode of birth (vaginal vs. caesarean), feeding method (breast feeding vs. infant formula), exposure to antibiotics, hygiene standards and geographical location [8,9]. It has been shown that the gut microbiota of infants who are vaginally delivered predominantly consist of *Bifidobacterium* in the weeks and months after birth, whereas those infants born via caesarean section predominantly consist initially of maternal skin microbiota, mainly *Staphylococcus* [10]. As the infant ages, the initial dominant aerobic gut environment matures and evolves to form an anaerobic environment, resulting in greater abundances of *Bifidobacterium* and Clostridia [7]. The gut microbiota begin to stabilise and evolve toward an adult-like composition by the second year of life [10].

The gut microbiota perform essential metabolic functions, serve as a source of essential nutrients and vitamins and assist in energy and nutrient extraction from food [11,12]. The gut microbiota also work in conjunction with the host’s defence and immune systems to protect against pathogen infiltration and colonisation. An imbalance in gut microbial composition can result in these mechanisms becoming disrupted and can contribute to increased intestinal permeability or a ‘leaky gut’ [13]. In the leaky gut, the tight junctions which support the epithelial lining barrier are disrupted, allowing infiltration of microbes, toxins or antigens into the tissues beneath. This may trigger inflammation through the activation of local and systemic immune responses. Shifts in the composition of the gut microbiota disrupt homeostasis and have been associated with long-term consequences, leading to disorders including CVD. 16S rRNA gene sequencing and whole-metagenome shotgun sequencing have added to the growing body of evidence that suggests that alterations of the gut microbiome composition affect the pathogenesis of CVD [7,14,15,16]. In addition to the composition of the gut microbiota, metabolites produced by these microbes can enter into the circulation and act as a modifiers of gut microbial effects on the host [17,18]. A number of microbiome studies have shown a potential link between microbial metabolites including short-chain fatty acids (SCFAs), bile acids (BA) and trimethylamine N-oxide (TMAO) and CVD [19,20,21].

The gut microbiota are involved in the digestion of food through two catabolic pathways, the saccharolytic pathway and the proteolytic pathway [22]. The saccharolytic pathway uses the gut microbiota to help break down sugars in the ingested foods, leading to SCFA production. The proteolytic pathway is characterised by the fermentation of ingested proteins, leading to SCFA production as well as formation of other metabolites including ammonia, thiols, amines, phenols and indoles. Diet is an important risk factor for CVD and the complex interactions between food and gut microbiota, and resulting metabolites play a key role in cardiovascular health. Given the growing body of evidence linking the gut microbiota with development of CVD, therapies aimed at modulating the gut microbiota have been suggested among the most promising strategies to prevent CVD.

## 2. Gut Microbiota and CVD

Many metabolites in the human body originate from intestinal microbes and there is growing evidence supporting the relationship between these metabolites and the development of CVD [18,23]. Through these metabolites, the gut microbiota influence host metabolic pathways such as cholesterol metabolism, in addition to inflammatory reactions and oxidative stress. Gut microbe-dependent metabolites may be absorbed into the bloodstream through the intestinal epithelium, which in turn influences the function of several organs and bodily systems [24]. Production of TMAO by the gut microbiota is a key mechanism of CVD. Dietary choline, phosphatidylcholine and carnitine are metabolised by gut microbes to produce trimethylamine (TMA). TMA is transported to the liver and further converted to TMAO via hepatic enzymes, flavin-containing monooxygenases [25,26]. TMAO has been shown to modulate cholesterol and sterol metabolism, cholesterol transport and bile acids levels [19,20,21,27]. Elevated serum levels of TMAO are associated with early atherosclerosis, severity of peripheral artery disease and with high risk of CVD mortality [28,29]. In a study of 4007 patients undergoing elective coronary angiography, increased plasma levels of TMAO were associated with an increased risk of death, MI, and stroke during three years of follow-up [21]. Dietary supplementation of TMAO to mice reduced the expression of Cyp7a1, the main bile acid synthetic enzyme and rate-limiting step in the catabolism of cholesterol, resulting in significantly reduced cholesterol transport [19]. In a study by Wang et al., associations were observed between CAD, peripheral vascular disease and MI and the gut metabolites choline, TMAO and betaine [20]. A TMAO-enriched diet expanded the atherosclerotic lesion area in atherosclerosis-prone mice (C57BL/6J Apoe−/−), while suppression of gut microbiota in atherosclerosis-prone mice inhibited dietary-choline-enhanced atherosclerosis. More recently, a murine study showed that both healthy and unhealthy diets can increase plasma levels of TMAO, with the gut itself being a significant site for oxidative production of TMAO [30]. Importantly this study reconciles some contradictory data on TMAO by showing no direct association between plasma TMAO and atherosclerosis extent but a clear association between TMAO levels and atherosclerotic plaque instability. This suggests that TMAO is a marker of cardiovascular risk.

SCFAs are the main metabolites produced by the gut microbiota through the fermentation of indigestible complex polysaccharides and proteins in the large intestine. The most abundant SCFAs are acetate (C2), propionate (C3), and butyrate (C4). SCFAs play important roles in gut health through maintenance of intestinal epithelial barrier integrity, reduction in inflammation, mucus production, and directly affecting immune cells. SCFAs bind to G protein-coupled receptors, the best-studied of which are free fatty acid receptors FFAR2 and FFAR3. FFAR2 is an essential receptor for neutrophil recruitment and through binding with FFAR2, SCFAs can help mediate inflammation [31]. SCFAs can induce satiety by stimulating the secretion of the gut hormones glucagon-like peptide-1 (GLP-1) and peptide YY (PYY) [32,33]. SCFAs can also influence blood pressure (BP) regulation by binding with FFAR2, FFAR3 and olfactory receptor 78 (OLFR78) [34]. Stimulation of OLFR78 has been demonstrated to elevate BP, while stimulation of FFAR3 lowered BP [35]. SCFAs regulate cholesterol distribution in liver and blood by inhibiting liver adipose synthetase activity. As SCFAs play a role in reducing serum cholesterol levels, decreases in SCFA production have been associated with dyslipidaemia. Concentrations of SFCAs have also been observed to be lower in patients with atherosclerotic vascular disease or hypertension [36].

The conversion of primary BAs to secondary BAs is also dependent on the gut microbiota. The gut microbiota produce bile salt hydrolase which catalyses the deconjugation of conjugated BAs and the release of free BAs and amino acids. A high intake of saturated fat leads to increased BA secretion and the presence of excess BAs in the intestine. Primary BAs are converted to hydrophobic secondary BAs, such as deoxycholic acid (DCA). Secondary BAs have been linked with increased hydrophobicity and enhanced membrane lipid binding and may cause changes in gut microbiota composition and structure [37,38]. Secondary BAs that enter the circulation can act as hormones and regulate signalling pathways including metabolism, inflammation and energy expenditure. Bile acid secretion disorders can cause blood lipid abnormalities and have been implicated in the development of atherosclerosis and other cardiometabolic diseases [3,38].

Components from the gut microbiota such as lipopolysaccharide (LPS) and peptidoglycan can trigger numerous immunomodulatory functions. LPS, a component of the Gram-negative bacterial cell membrane, is an endotoxin which promotes chronic systemic inflammation. LPS induces expression of inflammatory mediators and activates an immune response through Toll-like receptors (TLRs) and the nuclear factor kappa-light-chain-enhancer of activated B cells (NF-ĸB) pathway [39]. LPS may enter the circulation via intestinal absorption leading to endotoxemia, elevated levels of endotoxins in the blood. Endotoxemia has been associated with the pathogenesis of several disorders including atherosclerosis, obesity and insulin resistance [40]. In particular, endotoxemia has been shown to be associated with an increased risk of developing atherosclerosis in smokers and subjects with infectious disease [41].

The gut microbiota may also influence CVD development via oxidative stress. High oxygen levels lead to the formation of reactive oxygen species (ROS) including superoxide anion radicals, hydroxyl radicals, and hydrogen peroxide. During oxidative stress, ROS cause free-radical chain reactions, resulting in damage to proteins, DNA, and lipids, and even cell death [42]. Oxidative stress has been linked to several CVD risk factors including obesity, hypertension dyslipidaemia and diabetes. Endothelial dysfunction is a key hallmark underlying CVD and nitric oxide (NO) is the major factor accounting for endothelium-dependent relaxation in the aorta [43]. Increased ROS bioavailability reduces NO bioavailability and promotes inflammation, thereby affecting endothelial function [43]. Diets rich in foods with a high content of antioxidants and polyphenols have been shown to decrease oxidative stress markers. Elevated levels of glucose lead to an increase in oxidative stress, resulting in the oxidation of lipoproteins and production of adhesion molecules [44]. Gut microbes can influence oxidative stress through the regulation of uric acid metabolism. Elevated plasma uric acid levels may lead to an increase in the production of oxygen free radicals and promote oxidative stress and endothelial dysfunction [45].

## 3. Gut Microbiota Composition in Cardiovascular Disease

Next-generation sequencing (NGS) technology has greatly enhanced our understanding of the human gut microbiome. In particular, 16S rRNA gene sequencing has been extensively employed to profile the gut microbiota composition. The 16S rRNA gene is highly conserved between bacteria and contains nine hypervariable regions with bacterial species-specific variations. Alterations in the intestinal microbiota composition are believed to contribute to the pathogenesis of many diseases including CVD. Table 1 summarises the main findings of studies examining the relationship between the gut microbiota and CVD. Kelly et al. examined the association between the gut microbiota and lifetime CVD risk [46]. *Prevotella 2, Prevotella 7, Tyzzerella* and *Tyzzerella 4* were found to be enriched in subjects with high CVD risk compared with low risk subjects. By contrast, *Alloprevotella* and *Catenibacterium* were depleted among subjects with high versus low CVD risk. Alpha diversity was inversely associated with lifetime CVD risk.

The gut microbiota have been suggested to be an important contributor to the development of obesity, a key risk factor for CVD. Bäckhed et al. initially proposed that the gut microbiota was an important environmental factor that affects predisposition towards energy storage and fat mass gain [47]. Conventional mice were observed to have a higher body fat content and higher metabolic rate than germ-free (GF) counterparts. The same group sought to examine the mechanisms underlying the protection against obesity in GF mice [48]. Here, they demonstrated that in contrast to conventional mice, GF mice were resistant to diet-induced obesity, even if the mice ingested the same amount of calories. The lean phenotype of GF mice was associated with mechanisms that result in increased fatty acid metabolism. Ley et al. reported a reduced abundance of Bacteroidetes and a proportionally increased abundance of Firmicutes in obese subjects [49]. An increased Bacteroidetes/Firmicutes ratio has been replicated in numerous studies analysing the microbiota composition in obese subjects [50,51,52,53]. However, in contrast to these findings, there have been many studies which have not observed an increase in the Bacteroidetes/Firmicutes ratio and in some cases have reported a decreased Bacteroidetes/Firmicutes ratio [54,55,56,57,58,59,60]. Obese subjects have also been shown to present with reduced microbial diversity compared with lean controls [61,62,63]. Several studies have reported that *Akkermansia muciniphila*, a mucin-degrading bacterium, is inversely correlated with obesity [64,65,66,67,68]. Ozato et al. reported an inverse association between the genus *Blautia* and visceral fat accumulation in Japanese adults [69]. Indeed, *Blautia* was the only genus significantly and inversely associated with visceral fat accumulation. A 2020 study reported a positive association between *Acidaminococcus* and body mass index (BMI), as well as waist and hip circumferences in Bangladeshi adults [70]. Companys et al. demonstrated that anthropometric parameters were positively associated with *Collinsella aerofaciens*, *Dorea formicigenerans* and *Dorea longicatena*, which had higher abundances in overweight/obese subjects [71]. Saturated fatty acids were negatively associated with the *Intestinimonas*, a biomarker of the overweight/obese subjects. By contrast, saturated fatty acids were positively associated with *Roseburia*, a biomarker for the lean subjects.

The gut microbiota also have the capacity to contribute to dyslipidaemia. Martinez-Guryn et al. showed that GF mice exhibit decreased plasma triglyceride and LDL-C compared to control mice [72]. Karlsson et al. reported a positive correlation between high-density lipoprotein cholesterol (HDL-C) and *Clostridium* species and a negative correlation between serum triglycerides and *Clostridum* species [73]. Rebolledo et al. assessed differences in gut microbial composition between hypercholesterolemic subjects and controls [74]. Hypercholesterolemic subjects harboured a lower richness and diversity of bacterial communities than controls. Fu et al. examined the gut microbiota composition in 893 subjects from the LifeLines-DEEP population cohort, a Dutch population-based study [75]. Thirty-four bacterial taxa were associated with BMI and blood lipids. In particular, *Eggerthella* was associated with increased triglyceride and decreased HDL-C, and family Pasteurellaceae with decreased triglyceride. The authors reported that the microbiome had little effect on LDL-C or total cholesterol (TC). A more recent 2021 study showed that genera of the Lachnospiraceae family are negatively associated with lipid CVD risk factors including body fat, LDL-C and TC [71]. Liang et al. demonstrated that a high-fat and high-cholesterol diet (HFHCD) significantly decreased the alpha diversity of the gut microbiota in mice compared with high-fat diet (HFD) alone [76]. In addition, HFHCD mice had reduced abundances of *Oscillospira, Odoribacter, Bacteroides*, and *Prevotella* compared with HFD and greater *Ruminococcus* and *Akkermansia*. In 2021, Zhang et al. reported that mice fed a HFHCD had increased *Mucispirillum, Desulfovibrio, Anaerotruncus* and Desulfovibrionaceae, while *Bacteroides* and *Bifidobacterium* were depleted [77]. This finding was further corroborated in human hypercholesteremia patients.

With regard to CAD, Emoto et al. reported that the phylum Bacteroidetes was decreased and the order Lactobacillales was increased in CAD patients compared with controls [78]. Mitra et al. demonstrated differences in microbial composition between symptomatic and asymptomatic atherosclerotic plaques [79]. Asymptomatic atherosclerotic plaques contained increased abundances of Porphyromonadaceae, Bacteroidaceae, Micrococcacaea, and Streptococcacaea. In contrast, symptomatic plaques had an increased abundance of Helicobacteracaea, Neisseriaceae, and Thiotrichacaea. However, it remains unclear whether these bacteria are active or passive participants in plaque erosion/rupture. *Collinsella* was found to be enriched in patients with symptomatic atherosclerosis, while *Roseburia* and *Eubacterium* were enriched in healthy controls [80]. More recently, in a study by Zhu et al., decreased diversity and richness were observed in CAD patients [81]. *Escherichia-Shigella* were significantly enriched, while *Faecalibacterium, Subdoligranulum, Roseburia* and *Eubacterium rectale* were significantly reduced in CAD patients. Jie et al. showed an increase in the relative abundances of *Escherichia coli, Enterobacter aerogenes, Klebsiella* and *Streptococcus* in faecal samples from patients with atherosclerotic CVD than in healthy controls [82]. The relative abundances of bacteria typically associated with the oral cavity, such as *Streptococcus, Lactobacillus salivarius, Solobacterium moorei*, and *Atopobium parvulum*, were also higher in atherosclerotic patients than in healthy controls. Periodontal disease has been associated with atherosclerosis, suggesting that oral pathogens may contribute to the pathogenesis of atherosclerosis [83]. Koren et al. compared the microbial composition of oral, gut, and atherosclerotic plaque samples in patients with atherosclerosis [84]. *Chryseomonas* was detected in all atherosclerotic plaque samples. A correlation was noted between the combined abundances of *Veillonella* and *Streptococcus* in atherosclerotic plaques and in samples from the oral cavity. In addition, several OTUs were shared between the atherosclerotic plaque and oral or gut samples within the same individual. Hyvarinen et al. investigated the association between CAD and salivary levels of four major periodontal pathogens, *Aggregatibacter actinomycetemcomitans, Porphyromonas gingivalis, Prevotella intermedia*, and *Tannerella forsythia* [85]. High salivary levels of *A. actinomycetemcomitans* were associated with an increased risk for both acute and stable CAD.

A number of studies have investigated whether changes in intestinal microbiota composition affect hypertension. Moghadamrad et al. demonstrated that GF mice presented with significantly lower BP when compared with conventional mice, suggesting a contribution of the intestinal microbiota to the regulation of hypertension [86]. An increased Firmicutes/Bacteroidetes ratio in hypertensive animal models and patients has been widely reported [87,88]. Callejo et al. analysed the faecal microbiota composition in a rat model of pulmonary arterial hypertension (PAH) using 16S rRNA sequencing [89]. No differences in alpha and beta diversities were observed. However, PAH rats had a significantly increased Firmicutes/Bacteroidetes ratio. A 2015 study examined the gut microbiota composition in both animal and human hypertension [87]. Microbial richness, diversity, and evenness were observed to be significantly decreased in the spontaneously hypertensive rat (SHR) and in a chronic angiotensin II infusion rat model. An increased Firmicutes/Bacteroidetes ratio and decreases in acetate- and butyrate-producing bacteria were also observed. The microbiota of the human hypertensive subjects was observed to have reduced diversity compared with that of control subjects. Li et al. found decreased microbial richness and diversity, and an increase in *Prevotella* and *Klebsiella* in both pre-hypertensive and hypertensive populations compared to healthy controls [90]. A 2020 study used a multivariable-adjusted model to examine the association between gut microbiota composition and BP in 6953 subjects in Finland [91]. The associations between overall microbial composition and BP were reported as weak. However, mainly positive significant associations between BP indexes and 45 genera (27 belonging to the phylum Firmicutes) were observed. Strong negative associations between 19 *Lactobacillus* species and BP were also found. In particular, *Lactobacillus paracasei*, a well-studied probiotic, was associated with lower mean arterial pressure and lower dietary sodium intake.

## 4. Gut Microbiota Function in Cardiovascular Disease

In contrast to 16S compositional analysis, metagenomic approaches facilitate the comprehensive analysis of microbial community functional potential as well as strain-level analysis. The gut microbiota are believed to encode at least 150-fold more genes than the human genome [92,93]. Metabolomic analysis has become an important complementary approach in microbiome studies, allowing researchers to directly measure the metabolites present in the intestine. Many metabolites in the human body originate from intestinal microbes and there is growing evidence supporting the relationship between these metabolites and the development of CVD [18,23].

A 2021 study identified high-fat diet (HFD)-induced alterations of 19 metabolites in the liver and 43 metabolites in cecal contents of mice as potential biomarkers related to obesity [94]. HFD was associated with increased lipid profiles and total bile acid. Hong et al. studied the gut functional and metabolite profile in C57BL/6J mice that were fed with chow or HFD with or without Astragalus polysaccharides supplementation (APS) [95]. HFD was associated with an increase in six bacterial pathways, while eight pathways were reduced. These included tryptophan metabolism and methane metabolism, which were significantly enriched by HFD and reversed by APS supplementation. In addition, the nicotinate and nicotinamide metabolism pathway was reduced by HFD intake and increased by APS. Purine metabolism and glutathione metabolism were reduced and increased by HFD, respectively. Twenty species, including *Streptococcus equi* and *Bizionia argentinensis*, were negatively correlated with glutamic acid and pyroglutamic acid. Medina et al. analysed microbial functional patterns in obese and lean people from six different regions and observed that microbial contribution to functional pathways were region specific [96]. In addition, the authors analysed functional changes in the gut microbiota of obese patients after bariatric surgery and reported that changes were specific to the type of obesity treatment received. In 2019, Tran et al. used a metaproteomic approach to identify microbial factors that may correlate to diet induced obesity in mice prior to and following administration of an obesogenic high-fat low-fibre diet for 8 weeks [97]. Large shifts in the overall structure of the metaproteome were observed following HFD, which appeared to be driven by proteins derived from Clostridiales and Bacteroidales. Functionally a decreased abundance of flagellin proteins was observed following HFD.

Karlsson et al. utilised shotgun sequencing to characterise the functional capacity in patients with symptomatic atherosclerosis. The gut microbiome of symptomatic atherosclerosis was enriched with proinflammatory peptidoglycan synthesis encoding genes and showed reduced production of anti-inflammatory β-carotenes compared with healthy controls [80]. Jie et al. performed a metagenome-wide association study on 218 patients with atherosclerotic CVD and 187 healthy controls [82]. The atherosclerotic patients’ faecal metagenome differed from that of the controls in the potential for transport or metabolism of several molecules associated with cardiovascular health. Genes required for the formation of TMA and the O-antigen of LPS were enriched in atherosclerotic samples. Zhu et al. utilised PICRUSt to predict bacterial functional potential based on 16S gene sequencing profiles from 70 patients with CAD and 98 healthy controls [81]. Genes related to the phosphotransferase system, propanoate metabolism, LPS biosynthesis, and amino acid metabolism were increased in CAD patients, indicative of the inflammation observed in CAD. Feng et al. utilised both metagenomic and metabolomic approaches to identify microbial metabolites associated with CHD from plasma and urine of CHD patients and healthy controls [98]. Several metabolites were identified as novel candidate biomarkers, including GlcNAc-6-P, mannitol and 15 plasma cholines. Spearman correlation analysis of those biomarkers with microbial species showed that GlcNAc-6-P and mannitol were positively associated with *Clostridium* sp. HGF2, *Streptococcus* sp. M334 and *Streptococcus* sp. M143. Zhou et al. characterised the blood microbiome in a cohort of 199 subjects with CHD, ST-segment elevation myocardial infarction (STEMI) or healthy controls using shotgun sequencing [99]. In STEMI patients, higher microbial richness and diversity were observed. *Lactobacillus, Bacteroides*, and *Streptococcus* were found in more than 12% of post-STEMI blood samples. In comparison to control and CHD patients, STEMI patients’ LPS and d-lactate levels increased and correlated with systemic inflammation and predicted adverse cardiovascular events.

Kim et al. performed shotgun metagenomics on faecal samples from patients with high BP and healthy controls [100]. In particular, altered butyrate production in patients with high BP was observed. Significant increases in plasma biomarkers of gut epithelial barrier dysfunction and altered immune status including intestinal fatty acid-binding protein (I-FABP), LPS, and augmented gut-targeting proinflammatory T helper 17 (Th17) were also observed in high BP patients. In addition, Zonulin, a gut epithelial tight junction protein regulator, was elevated in high BP patients and strongly correlated with systolic blood pressure (SBP). A 2017 metagenomic and metabolomic analyses of 56 subjects with pre-hypertension, 99 individuals with primary hypertension and 41 healthy controls, reported microbial functional alterations in the gut microbiota of both pre-hypertensive and hypertensive patients [90]. The metabolic functions and metabolites identified in pre-hypertensive and hypertensive adults were closely linked to inflammation. Wang et al. characterised microbial composition and function in two subtypes of hypertension; isolated systolic hypertension (ISH) and isolated diastolic hypertension (IDH) [101]. A reduction in bacterial richness and evenness was reported in IDH patients compared with ISH and healthy controls. Patients with IDH or ISH had increased abundances of *Rothia mucilaginosa* and reduced *Clostridium* sp. *ASBs410* compared to controls. Functionally, KEGG modules such as sodium transport system were significantly decreased in IDH patients compared with control. In ISH patients, functions related to biotin biosynthesis were decreased. A recent 2021 study used whole-metagenome shotgun sequencing to analyse stool samples from patients diagnosed with both depression and hypertension (DEP-HTN) [102]. The endotype of DEP-HTN patients was distinctive from DEP or HTN alone and defined by co-occurrence of *Eubacterium siraeum, Alistipes obesi, Holdemania filiformis*, and Lachnospiraceae bacterium 1.1.57FAA with *Streptococcus salivariu*. Metagenomes of DEP-HTN patients engaged pathways that degrade beneficial SCFAs and gamma amino butyric acid (GABA) and were associated with enhanced inflammasome activities and sodium absorption.

Hayashi et al. reported an increase in TMAO levels in patients with both compensated and decompensated heart failure in addition to an altered gut microbiota composition [103]. In 2021, the same researchers used metagenome-wide shotgun sequencing to elucidate which of the key genes of the TMA synthesis pathways contributes to the increase in TMAO levels [104]. The abundance of cntA/B, derived mainly from *Escherichia* and *Klebsiella* was positively correlated with TMAO, in heart failure patients. No correlation was observed for cutC/D or betaine reductase. Cui et al. performed metagenomics and metabolomic analyses of faecal and plasma samples from patients with chronic heart failure (CHF) and controls [105]. A decreased abundance of *Faecalibacterium prausnitzii* and an increase in *Ruminococcus gnavus* were observed in CHF patients. Microbial genes involved in the metabolism of protective metabolites such as butyrate were significantly reduced in the metagenome of CHF patients. Meanwhile, genes for potentially harmful metabolites such as TMAO were significantly upregulated in CHF patients.

## 5. Gut Microbiota as Therapeutic Strategies for Cardiovascular Disease

### 5.1. Probiotics

Probiotics are defined by the FAO/WHO as “live microorganisms which, when administered in adequate amounts, confer a health benefit on the host” [106]. A growing body of evidence demonstrates the beneficial health impacts of probiotic supplementation and they could be considered as a potential therapeutic tool for CVD. Probiotics have been investigated in several studies for their use as anti-obesity treatments. Pedret et al. examined the effect of *Bifidobacterium animalis* subsp. *lactis* CECT 8145 (Ba8145) on 135 obese subjects. Ba8145 decreased BMI when compared with a baseline and placebo group and was associated with an increase in *Akkermansia*. A three-week RCT involving 25 obese hypertensive participants investigated if a hypocaloric diet (1500 kcals/day) supplemented with a probiotic cheese containing the *Lactiplantibacillus plantarum* strain TENSIA could reduce symptoms of metabolic syndrome (MetS) [107]. BMI was reported to be significantly reduced in the participants that received probiotic cheese versus the control group. A 12 week RCT randomised 210 healthy Japanese adults with large amounts of visceral fat to receive fermented milk containing *Lactobacillus gasseri* SBT2055 (LG2055) [108]. Participants who received LG2055 exhibited a significant lowering in abdominal visceral fat, waist and hip circumference, BMI and body fat mass when compared with those in the control group.

Another RCT investigated the impact of daily supplementation of *Lactobacillus rhamnosus* CGMCC1.3724 (LPR) with oligofructose and inulin for 24 weeks in 125 obese adults combined with an energy-restricted diet for the first 12 weeks [109]. LPR supplementation did not significantly affect weight loss compared to placebo. However, a significant reduction in body weight was observed in the female participants as a result of dietary intervention compared with the placebo. Cai et al. reported that *L. plantarum* FRT10 supplementation alleviated HFD-induced obesity in mice partly related to the activation of the peroxisome proliferator-activated receptor-α (PPARα)/carnitine palmitoyltransferase-1α (CPT1α) pathway [110]. FRT10 reduced fat weight, body weight gain, liver triacylglycerols and alanine aminotransferase concentrations. Supplementation with FRT10 was also associated with significantly increased abundances of *Butyricicoccus, Butyricimonas, Intestinimonas, Odoribacter, and Alistipes*, and decreased abundances of Desulfovibrionaceae, *Roseburia*, and *Lachnoclostridium*. Depommier et al. conducted a randomised controlled pilot study in 32 overweight/obese insulin-resistant individuals [111]. Daily supplementation of *A. muciniphila* significantly improved insulin sensitivity and reduced insulinemia and plasma TC compared to placebo.

Probiotic bacteria have been studied as a therapeutic to improve lipid profiles. Probiotic bacteria with bile salt hydrolase activity may reduce serum cholesterol levels through deconjugation of BAs and increasing their excretion [112,113]. This increases the need for de novo bile synthesis, leading to increased conversion of cholesterol into BAs in the liver, thus reducing serum cholesterol levels. SCFAs have been shown to regulate cholesterol metabolism and reduce hepatic cholesterol synthesis [114]. Other potential mechanisms of action include probiotic-driven cholesterol assimilation and increasing cholesterol binding to cell walls of probiotics in the small intestine, thereby decreasing the amount absorbed by the body [115]. A 2021 study showed that *L. plantarum* K50 supplementation in HFD-induced obese mice decreased the serum triglyceride level and increased HDL-C levels [116]. Ahn et al. evaluated the triglyceride-lowering effect of supplementation with *Lactobacillus curvatus* HY7601 and *L. plantarum* KY1032. The RCT was performed over 12 weeks and involved 92 participants with hypertriglyceridaemia. Probiotic supplementation resulted in a significant 20% reduction in serum triglycerides and a 25% increase in apo A-V. Another RCT studied the effect of a fermented soy product containing *Enterococcus faecium* CRL 183 and *Lactobacillus helveticus* 416 and isoflavone on CVD risk markers in moderately hypercholesteraemic men for 42 days [117]. Consumption of the probiotic soy product with isoflavone improved total cholesterol (TC), LDL-C and electronegative LDL concentrations. The levels of high-density lipoprotein-cholesterol (HDL-C) was unaffected and the level of fibrinogen and C-reactive protein (CRP) levels were not improved. The effect of a combination of *Lactobacillus acidophilus* and *Bifidobacterium bifidum* or placebo for six weeks on lowering serum cholesterol in 70 hypercholesteraemic patients was evaluated [118]. The probiotic group exhibited decreases in TC, HDL-C and LDL-C levels compared with the control at the end of treatment. We previously demonstrated the potential of exopolysaccharide (EPS)-producing *Lactobacillus mucosae* DPC 6426 as hypocholesterolaemia therapy in apo E deficient (apoE−/−) mice, fed a high fat/high cholesterol diet [119]. In another study, we fed apoE−/− mice either a high-fat/cholesterol diet alone or in conjunction with plant sterol ester, oat β-glucan (OBG), BSH active *Lactobacillus reuteri* APC 2587, or the drug atorvastatin [120]. OBG intervention resulted in the most favourable shifts in microbiome composition and functionality, and OBG mice appeared to be protected from the high-fat/cholesterol-induced atherogenesis.

Pregnancy is associated with elevated levels of lipid concentrations. In a RCT conducted by Hoppu et al., 256 pregnant women were randomised into three study groups: dietary counselling with probiotics (*L. rhamnosus* GG and *Bifidobacterium lactis*) or dietary counselling with placebo or control in the first trimester of pregnancy [121]. Similar lipid levels were observed in all groups during pregnancy. However, postpartum LDL-C and TC were lower in both the probiotic and placebo dietary counselling groups compared with controls. Another RCT examined the effects of daily supplementation of a probiotic yoghurt containing *Streptococcus thermophilus*, *Lactobacillus bulgaricus*, *L. acidophilus* LA-5, and *Bifidobacterium animalis* BB12 among 70 pregnant women in the third trimester for 9 weeks [122]. Consumption of the probiotic yoghurt resulted in a significant reduction in TC, LDL-C and HDL-C levels, as well as serum triglyceride concentration. However, no significant differences were found in serum lipid profiles.

Probiotics have also been studied for their putative antihypertensive effects. Probiotics may exert this antihypertensive effect through reducing blood glucose levels, insulin resistance and by improving cholesterol levels and endothelial dysfunction [123,124]. Seppo et al. evaluated the BP-lowering effect of milk fermented by *L. helveticus* LBK-16H in 39 hypertensive subjects [125]. *L. helveticus* LBK-16H supplementation resulted in decreases in both SBP and diastolic blood pressure (DBP) compared with control group. Ivey et al. examined the effect of *L. acidophilus* La5 and *B. lactis* Bb12 on BP in 156 overweight men and women over 55 years [126]. When compared to control, probiotic yoghurt did not significantly alter BP, heart rate or serum lipid concentrations. A 2015 RCT examined the effects of probiotic soy milk containing *L. plantarum* A7 on BP in type 2 diabetic (T2D) patients [127]. Probiotic soymilk was reported to significantly decrease SBP and DBP. Aihara et al. demonstrated that daily consumption of fermented milk with *L. helveticus* CM4 in subjects with high–normal BP or mild hypertension decreased SBP and DBP without any adverse effects [128]. In a 2020 study, administration of *L. plantarum* WJL during pregnancy and lactation in dams was reported to reduce BP and prevent cardiovascular dysfunction in male offspring [129].

The potential for probiotic bacteria to decrease ROS and oxidative stress has also been examined. Probiotics may exert antioxidant effects through regulation of antioxidant enzymes, scavenging of ROS or chelating of metal ions [42]. LeBlanc et al. demonstrated that mice receiving catalase or superoxide dismutase-producing *Lactobacillus casei* BL23 had a faster recovery of initial weight loss, increased enzymatic activities in the gut and less colonic inflammation compared to animals that received wild-type strain [130]. Ejtahed et al. assessed the effect of a probiotic yogurt containing *L. acidophilus* La5 and *B. lactis* Bb12 on blood glucose and antioxidant status in 64 T2D patients [131]. Subjects receiving probiotic yogurt had significantly increased erythrocyte superoxide dismutase and glutathione peroxidase activities and total antioxidant status compared with the control group. Gómez-Guzmán et al. demonstrated that administration of *Lactobacillus fermentum* CECT5716, or *Lactobacillus coryniformis* CECT5711 plus *Lactobacillus gasseri* CECT5714 improved endothelial dysfunction, vascular inflammation and vascular oxidative stress, in SHR. Xin et al. reported that supplementation of *Lactobacillus johnsonii* BS15 alleviated HFD-induced oxidative stress in non-alcoholic fatty liver disease (NAFLD) mice, suggesting that this strain can improve host redox state [132]. Friques et al. evaluated the effects of kefir treatment for 60 days on endothelial function and vascular responsiveness in SHR [133]. Kefir was reported to attenuate increased ROS production and decreased nitric oxide NO bioavailability in SHR, thereby improving endothelial function.

Malik et al. examined whether oral *L. plantarum* 299v supplementation for six weeks in 24 men with stable CAD improves vascular endothelial function and decreases systematic inflammation [128]. Lp299v supplementation resulted in improvements in endothelium-dependent vasodilation and increased NO bioavailability. In addition, a decrease in systemic inflammation with reduced IL-8, IL-12 and leptin levels was observed. It was also found that the SCFA propionate increased with Lp299v supplementation. A 12 week RCT examined the effect of a multispecies probiotic, Ecologic^®^ Barrier, on endothelial dysfunction in obese postmenopausal women [134]. The probiotic preparation contained *Bifidobacterium bifidum* W23, *B. lactis* W51, *B. lactis* W52, *L. acidophilus* W37, *Lactobacillus brevis* W63, *Lactobacillus casei* W56, *Lactobacillus salivarius* W24, *Lactococcus lactis* W19, and *Lactococcus lactis* W58. Eighty-one women participated in the trial and were randomised to receive either a low dose of probiotic, a high dose or placebo daily. The trial revealed a significant reduction in SBP, vascular endothelial growth factor, interleukin-6 (IL-6), tumour necrosis factor alpha (TNF-α), and thrombomodulin following high-dose probiotic supplementation. Decreased SBP and IL-6 were observed with low doses of probiotic supplementation The anti-inflammatory effects of probiotics in relation to CVD were also examined in the 2019 PROSIR study. This study evaluated the effect of *L. reuteri* V3401 on gut microbiota composition, biomarkers of inflammation, CVD risk and hepatic steatosis in obese adults with MetS [127]. The RCT included 60 participants, aged 18 to 65 years, who were randomised to receive daily dose of *L. reuteri* V3401 or placebo for 12 weeks. Consumption of *L. reuteri* V3401 was associated with lower levels of inflammatory biomarkers IL-6 and soluble intercellular adhesion molecule-1 (sVCAM-1).

As previously discussed, TMAO has been reported to be a risk factor in the development of CVD. A 2015 RCT examined *L. casei* Shirota (LcS) supplementation on TMAO levels in 13 patients with MetS [135]. There were no significant differences observed in TMAO levels between subjects receiving LcS and controls. Tenore et al. tested the effects of lactofermented Annurca apple puree (lfAAP) containing *L. rhamnosus* LRH11 and *L. plantarum* SGL07 on plasma lipid profiles and TMAO levels in 90 individuals with CVD risk factors [136]. lfAAP supplementation significantly increased HDL-C levels and lowered TMAO levels at the end of the intervention period. Malik et al. reported that *L. plantarum* 299v supplementation in 20 men with stable CAD did not significantly change plasma TMAO concentrations [137]. Qiu et al. reported that *L. plantarum* ZDY04 significantly reduced serum TMAO and cecal TMA levels, as well as significantly inhibited the development of TMAO-induced atherosclerosis in ApoE−/− 1.3% choline-fed mice compared to controls [138]. Boutagy et al. investigated if supplementation of the multistrain probiotic VSL#3 could attenuate the increase in fasting plasma concentrations of TMAO in 19 healthy males following a HFD [139]. The authors reported that no differences in TMAO levels were observed between the VSL#3 arm and the control arm of the study. Matsumoto et al. investigated the effect of *B. lactis* LKM512 on colonic TMA in healthy volunteers [140]. Faecal TMA concentration and the proportion of TMA-producing bacteria were found to be lower in the probiotic group than in the placebo. Borges et al. evaluated the effects of *S. thermophilus* (KB19), *L. acidophilus* (KB27), and *Bifidobacterium longum* (KB31) supplementation on TMAO plasma levels in chronic kidney disease patients on haemodialysis [141]. Probiotic supplementation did not appear to change plasma TMAO levels. However, a significant increase in betaine plasma levels was observed.

A 2015 meta-analysis included 15 studies totalling 788 adults to examine the effect of probiotics on the reduction in CVD risk factors [142]. The authors reported a statistically significant pooled effect reduction in TC, LDL-C, BMI, waist circumference and inflammatory markers. A significant reduction in LDL-C was reported when *L. acidophilus* was used in a trial compared to other strains. Subgroup analysis revealed that probiotics were most effective on TC and LDL-C when consumed in the form of fermented milk or yoghurt compared to capsule form, consumption was at least eight weeks, and were multistrain rather than a single strain intervention. Mo et al. assessed the efficacy of probiotics in lowering serum lipid concentrations in a meta-analysis of 19 RCTs including 967 hypercholesterolemia adults [143]. The meta-analysis indicated that the use of probiotics can lower TC and LDL-C compared to controls. These effects were greatest for longer intervention times, certain probiotic strains, and in younger mildly hypercholesterolemia subjects. No significant effects on TG and HDL-C levels were observed. The effects of probiotics on decreasing TC and LDL-C levels were greater for longer intervention times and in younger mildly hypercholesterolemia subjects. With regard to strain specific effects, *L. acidophilus, L. plantarum* and *L. reuteri* significantly decreased TC. *L. plantarum, L. helveticus* and *E. faecium* significantly reduced LDL-C. No significant beneficial effects were observed for *L. fermentum, L. rhamnosus* and mixtures of *L. acidophilus* and *Bifidobacterium*.

A more recent and larger 2020 systematic review and meta-analysis examined the efficacy of probiotics on lowering CVD risk factors (obesity, high BP, high cholesterol and triglycerides, elevated HbA1c and serum glucose) in patients with CVD or co-morbidities, such as hypertension, obesity, type II diabetes, MetS and hypercholesterolemia [144]. Thirty-four RCTs with 2177 adults who received probiotic intervention (*n* = 1176) or control (1001) were included in the analysis. An overall statistically significant reduction in SBP and DBP, TC, LDL-C, serum glucose, HbA1c and BMI with probiotic intake was reported. Subgroup analysis revealed that the statistically significant effects of probiotics on the lowering of risk factors might be related to patient morbidity. Diabetic patients displayed the greatest reduction in SBP, DBP, triglycerides, HDL-C and fasting glucose. Hypercholesterolemia patients displayed the greatest reduction in TC and LDL-C, as did obese patients in reducing BMI. Female populations were reported to have more significant reduction in fasting glucose, TC, LDL-C, HbA1c and BMI compared to males. The authors hypothesise that the oestrogen-gut microbiome axis may contribute to this observed reduction in CVD risk factors. Concerning probiotic formulation, the use of probiotics in yoghurt was statistically favoured for a reduction in TC, HbA1c, fasting glucose and LDL-C. Kefir and powder formulations were associated with significant reductions in BMI. Results favoured a longer duration of >1.5 months for reductions in SBP and DBP, while shorter duration favoured significant reductions in TC, LDL-C HbA1c and fasting glucose. Dosage of probiotics was also important as higher daily doses (>1 × 10^9^ CFU) were reported to be more effective in reducing TC, BMI, HbA1c and fasting glucose. Limitations of this meta-analysis include small population sizes and the use of different probiotic strains and doses in the various studies.

### 5.2. Faecal Microbiota Transplantation

Faecal microbiota transplantation (FMT) is emerging as an innovative therapeutic solution for many microbiota-associated conditions. FMT involves the collection of filtered faecal samples from carefully screened health donors and its placement into the intestinal tract of the recipient with the aim of restoring the normal functions of the gut microbiota. Eisman et al. first described the use of FMT, via faecal enema, as an adjunct in the treatment of pseudomembranous colitis [145]. In particular, the use of FMT in the treatment of recurrent *Clostridium difficile* infection has proved extremely successful and has been demonstrated to be significantly more effective than using common treatment methods such as vancomycin antibiotic treatment [146,147]. FMT is also emerging as a potential therapeutic intervention for other microbiota-associated disorders including ulcerative colitis, Crohn’s disease, obesity, type 2 diabetes and CVD.

A pioneering study by Bäckhed et al. showed a 60% increase in body fat content and insulin resistance in GF mice inoculated with FMT from conventionally raised mice [47]. Kootte et al. studied the effect of lean donor versus autologous FMT to male recipients with MetS [148]. Lean donor FMT significantly improved insulin sensitivity accompanied by altered gut microbiota composition at six weeks. The FMT-TRIM pilot trial randomised 24 adults with obesity who were at high risk for development of T2D to receive either weekly oral FMT capsules from lean donors or placebo [149]. No significant improvements in insulin sensitivity or metabolic outcomes in the FMT group compared to the placebo group were observed. Limitations of this study were the small sample size, inclusion of subjects with relatively mild insulin resistance, and lack of concurrent dietary intervention. Allegretti et al. examined the effects of FMT with oral capsules from a lean donor in 22 obese subjects [150]. The microbiota composition and bile acid profiles of subjects who received FMT were observed to have shifted towards that of the donor.

Mell et al. used rat models of hypertension and normotension, Dahl salt-sensitive (S) and Dahl salt-resistant (R) rats, to examine whether differences in microbiota contribute to BP using caecal microbiota transplantation [151]. Caecal transplantation did not alter the SBP of the R rats. By contrast, SBP of the S rats given R rat microbiota was significantly elevated during the rest of their life, and they also had a shorter lifespan. Compositional analysis indicated that reduced faecal Veillonellaceae and increased plasma heptanoate and acetate were associated with the increased BP observed in the S rats given R rat caecal content. Furthermore, it has been demonstrated that transplantation of caecal contents from hypertensive obstructive sleep apnoea rats on HFD into recipient rats on normal chow diet lead to higher BP levels [152]. Recipient rats receiving caecal contents from a sham donor on a HFD had no change in BP. The findings suggest that the hypertensive phenotype of obstructive sleep apnoea rats is transferrable via the caecal contents. Li et al. also reported that high blood pressure could be transferred by FMT [90]. FMT from hypertensive human donors to GF mice resulted in elevated BP. Gregory et al. investigated if caecal microbial transplantation can transmit choline diet-induced TMAO production and atherosclerosis susceptibility [153]. An atherosclerosis-prone and high TMAO-producing model, C57BL/6J, and an atherosclerosis-resistant and low TMAO-producing model, NZW/LacJ, were used as donors into antibiotic treated Apoe−/− mouse mice. Enhanced atherosclerosis was observed in mice receiving caecal microbes from atherosclerosis-prone donors compared to those from atherosclerosis-resistant donors. The increase in atherosclerosis was only observed with a high choline diet. Differences in TMA and TMAO levels were initially reported in the post-transplantation period. However, this was not maintained to the end of the study.

In a study carried out by Hu et al., using an experimental autoimmune myocarditis (EAM) mouse model investigated the use of FMT for the treatment of myocarditis was investigated [154]. FMT treatment was shown to restore the Bacteroidetes population and reduce the incidence of myocarditis. Another preclinical study investigated if changes in the gut microbiota induced by reciprocal FMT from normotensive Wistar-Kyoto (WKY) to SHR could alter neuroinflammation and sympathetic activity and thereby lower BP [155]. The animals were divided into four groups: WKY transplanted with WKY microbiota, SHR with SHR, WKY with SHR, and SHR with WKY. Following FMT from WKY to SHR rats, basal SBP and DBP decreased, however, heart rate did not change significantly. An increase in basal SBP and DBP was also observed after FMT from SHR to WKY rats. Correlation analyses showed a negative correlation between the abundances of *Blautia* and *Odoribacter* and high SBP. Kim et al. demonstrated that FMT from healthy donor mice who were fed the polyphenol, Resveratrol, to hypertensive mice had beneficial effects in significantly lowering SBP as well as improving glucose homeostasis during diet-induced obesity [156]. Smits et al. investigated the effects of FMT from vegan donors on TMAO production and vascular inflammation in 20 obese male subjects suffering from MetS [157]. Dietary L-carnitine, which is converted by members of the intestinal microbiota to TMAO, is abundant in red meat, while, individuals with a vegan diet have lower amounts of circulating TMAO [19,158]. Vegan-donor FMT resulted in changes in the gut microbiota composition in some but not all recipients. Beneficial changes in TMAO production or parameters relating to vascular inflammation were not observed.

## 6. Conclusions and Future Directions

Prevention of CVD is dependent on understanding the mechanisms underlying disease development. Increasing evidence points to an association between alterations in the composition of the gut microbiota and microbial metabolites with the development of CVD. Studies analysing microbial communities have identified several taxa associated with CVD development and have emphasised the importance of oral microbiota in CVD risk assessment and therapeutics. More conclusive evidence is still needed to determine whether alterations in gut microbiota composition are a consequence or a causal factor in the pathogenesis of CVD. Rather than focusing on individual taxa, profiling of microbial metabolic potential and functional genetic alterations in CVD through metagenomic and metabolomic approaches will aid our understanding of the influence of the gut microbiota on CVD pathogenesis. The combined use with other “omics” such as transcriptomics and proteomics will assist in the discovery of novel biomarkers and could act as a form of microbial fingerprinting, facilitating more targeted precision therapies. Therapeutic strategies focusing on the inhibition or blocking of microbial metabolic pathways such as the TMAO signalling pathway could have potential for CVD intervention. Probiotic interventions hold promise in CVD risk prevention, though larger multicentre trials, with information on doses, frequencies of administration and specific strains must be performed to generate more convincing data. The use of FMT in the treatment of CVD has shown some promise as a new therapeutic approach. However, much of the research has been performed in preclinical rather than clinical settings. Further limitations associated with the use of FMT include the possibility of the transfer of endotoxin and infectious agents, resulting in new complications. In conclusion, the gut microbiome plays an important role in the pathogenesis of CVD and targeting the gut microbiota with appropriate therapeutic interventions is a promising strategy for better CVD prevention and management in the future.

## Figures and Tables

**Table 1 metabolites-11-00493-t001:** Comparison of published studies of gut microbiome in CVD.

Study	Year	Cohort	Main Findings	PMID
Kelly et al.	2016	Bogalusa Heart Study subjects	*Prevotella*, *Tyzzerella* ↑ in subjects with high CVD risk. Alpha diversity inversely associated with CVD risk.	27507222
Bäckhed et al.	2004	Germ-free (GF) C57BL/6 mice	FMT from conventionalized mice ↑ body fat and insulin resistance.	15505215
Bäckhed et al.	2007	GF C57BL/6J mice	GF mice protected against diet-induced obesity through AMPK and Fiaf.	17210919
Ley at al.	2006	12 obese subjects	Obesity ↓ Bacteroidetes ↑ Firmicutes.	17183309
Ley at al.	2005	ob/ob mice, lean ob/+ mice	Obesity ↓ Bacteroidetes ↑ Firmicutes.	16033867
Turnbaugh et al.	2009	Obese and lean twin pairs	Core gut microbiome at level of metabolic functions.	19043404
Bervoets et al.	2013	26 obese and 27 lean children	Obesity ↑ Firmicutes/Bacteroidetes ratio.	23631345
Schwiertz et al.	2010	30 lean, 35 overweight and 33 obese subjects.	↑ *Bacteroides* from lean to obese.	19498350
Jumpertz et al.	2011	12 lean and 9 obese subjects	↑ Firmicutes ↓ Bacteroidetes in lean subjects with an increased energy harvest.	21543530
Zhang et al.	2009	3 normal weight, 3 morbidly obese, and 3 post-gastric-bypass subjects	Firmicutes dominant in normal-weight and obese subjects. ↓ Firmicutes ↑ Gammaproteobacteria in post-gastric-bypass.	19164560
Duncan et al.	2008	Obese and non-obese subjects	No evidence that Firmicutes/Bacteroidetes have a function in obesity.	18779823
Tims et al.	2013	40 monozygotic twin pairs	Inverse correlation between *Clostridium* cluster IV diversity and BMI. *Eubacterium ventriosum* and *Roseburia intestinalis* positively correlated to BMI differences. *Oscillospira guillermondii* negatively correlated to BMI differences.	23190729
Collado et al.	2008	18 overweight and 36 normal-weight pregnant women	Overweight ↑*Bacteroides* and *Staphylococcus*.	18842773
Kalliomäki et al.	2008	25 overweight and obese children, 24 normal-weight	↑ Bifidobacteria during infancy in normal-weight. ↑ *Staphylococcus aureus* during infancy in overweight.	18326589
Cotillard et al.	2013	38 obese and 11 overweight subjects	Dietary intervention improves low gene richness	23985875
Davis et al.	2017	81 random Alabama residents	Westernized diet type had a greater impact upon gut microbiota diversity than ↑ BMI	28677210
Stanislawski et al.	2019	- 152 obese and lean female adult twins and their mothers from Missouri - Lean subjects from Global Gut study - 5035 healthy subjects from American Gut cohort - 102 teenagers from the Exploring Perinatal Outcomes among Children study - 42 HIV positive and negative Mexican subjects	Gut microbiota phenotypes of obesity may differ with race/ethnicity	31285833
Palleja et al.	2016	13 morbidly obese patients who underwent Roux-en-Y gastric bypass (RYGB)	↑ microbial diversity and altered composition post RYGB	27306058
Schneeberger et al.	2015	C57BL/6J mice	Rapid decline of *Akkermansia muciniphila* during HFD and aging	26563823
Seck et al.	2018	1326 subjects with variable geo-graphical origin, diet, age, and gender	High fecal salinity linked to ↓diversity and ↓*Akkermansia muciniphila* and *Bifidobacterium*	30206336
Medina et al.	2017	9 obese subjects following medical dietary treatment, 5 following RYGB, 5 following sleeve gastrectomy	↑Proteobacteria in surgical patients	28649469
Zhou et al.	2020	10,534 subjects from American Gut Project	*Akkermansia* is associated with ↓risk of obesity	33110437
Ozato et al.	2019	1001 Japanese subjects	*Blautia* associated with visceral fat accumulation	31602309
Osborne et al.	2020	248 subjects from the Health Effects of Arsenic Longitudinal Study	Correlation between *Oscillospira* and leanness	31138061
Companys et al.	2021	96 overweight/obese subjects and 32 lean subjects	*Dorea formicigenerans, Dorea longicatena and Collinsella aerofaciens* obesity biomarkers. Lachnospiraceae associated with lipid CVD risk factors.	34199239
Martinez-Guryn et al.	2018	C57Bl/6 mice	HFD ↑ Clostridiaceae, ↓Bifidobacteriaceae and Bacteroidaceae GF mice ↓ plasma triglyceride and LDL-C	29649441
Karlsson et al.	2013	53 women who had Type 2 Diabetes, 49 with impaired glucose tolerance, 43 with normal glucose tolerance	Positive correlation between HDL-C and *Clostridium*, negative correlation between serum triglycerides and *Clostridum*	23719380
Rebolledo et al.	2017	30 hypercholesterolemia subjects, 27 normocholesterolemic controls	↓ microbial richness and diversity in hypercholesterolemic subjects.	28698878
Fu et al.	2015	893 subjects from the Life-Lines-DEEP cohort	34 taxa associated with BMI and blood lipids	26358192
Liang et al.	2021	C57BL/6 mice; normal chow, high-fat diet (HFD), or high-fat and high-cholesterol diet (HFHCD)	↑ *Oscillospira, Odoribacter, Bacteroides*, and *Prevotella*, ↓ Ruminococcus and *Akkermansia* in HFD compared with HFHCD	32999215
Zhang et al.	2021	- C57BL/6 mice; normal chow, HFLCD or HFHCD - 59 hypercholesterolemia subjects and 39 healthy controls	HFHCD associated with ↑ *Mucispirillum, Desulfovibrio, Anaerotruncus* and Desulfovibrionaceae; ↓ *Bifidobacterium* and *Bacteroides*	32694178
Emoto et al.	2016	39 coronary artery disease (CAD) patients, 30 controls with coronary risk factors and 50 healthy volunteers	CAD associated with ↑ Lactobacillales, ↓ Bacteroidetes	26947598
Mira et al.	2015	22 symptomatic and asymptomatic atherosclerotic plaques	Asymptomatic plaques ↑ Porphyromonadaceae, Bacteroidaceae, Micrococcaceae, and Streptococcaceae	26334731
Karlsson et al.	2012	12 patients with symptomatic atherosclerotic plaques	↑ *Collinsella* in patients with symptomatic atherosclerosis	23212374
Zhu et al.	2018	70 patients with CAD and 98 healthy controls	CAD associated with ↑ *Escherichia-Shigella, Enterococcus* ↓ *Faecalibacterium, Subdoligranulum, Roseburia, Eubacterium rectale*	30192713
Jie et al.	2017	218 individuals with atherosclerotic CVD and 187 healthy controls.	Atherosclerotic CVD associated with ↑ Enterobacteriaceae, *Streptococcus*	29018189
Koren et al.	2011	15 patients with atherosclerosis and healthy controls.	*Chryseomonas* in all atherosclerotic plaque samples, and *Veillonella* and *Streptococcus* in the majority.	20937873
Hyvärinen et al.	2012	179 patients with stable CAD, 166 with acute coronary syndrome, and 119 healthy controls.	↑ salivary levels of *A. actinomycetemcomitans* associated with ↑ risk for CAD.	22704805
Moghadamrad et al.	2015	GF male C57BL/6 mice, altered Schaedler flora (ASF) mice, and specific pathogen-free (SPF) mice	GF mice had significantly lower BP than conventional mice	25643846
Yang et al.	2015	- Wistar Kyoto (WKY) and spontaneously hypertensive (SHR) rats - Patients with normal and high SBP	SHR ↓ microbial richness, diversity, and evenness, ↑ Firmicutes/Bacteroidetes ratio	25870193
Marques et al.	2017	C57Bl/6 mice; control diet, high fiber or acetate	Fiber and acetate ↓ Firmicutes/Bacteroidetes ratio, ↑ *Bacteroides acidifaciens*	27927713
Callejo et al.	2018	Pulmonary arterial hypertension (PAH)-exposed wistar rats	PAH rats 3X ↑ Firmicutes/Bacteroidetes ratio.	29946072
Li et al.	2017	41 healthy controls, 56 pre-hypertension, 99 primary hypertension	Pre-hypertensive and hypertensive associated with ↓ microbial richness and diversity, ↑ *Prevotella* and *Klebsiella*	28143587
Palmu et al.	2020	6953 Finns aged 25 to 74 years	α and β diversities strongly related to BP indexes Positive, associations between BP indexes and 45 microbial genera, majority Firmicutes. Negative associations between 19 *Lactobacillus* species and BP indexes	32691653

## Data Availability

Not applicable.

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
