# Peer review of "Exploring the Gut Microbiota and Cardiovascular Disease"

_metabolites, 2021, doi:10.3390/metabo11080493_

Round 1

Reviewer 1 Report

This is a review article regarding cardiovascular animal models and gut microbiota. The authors described about animal models, from small to large, in the first half part. In the second half part, the authors described about gut microbiota; however, there was nothing to connect each other. This reviewer has some comment as described below.

Major comment:

  1. The authors should add “gut microbiota” issues in animal model, and then, they should go to human part.
  2. The authors should build a good flow from animal models to human gut microbiota. In current situation, there was no good association between animal and human.

Minor comment:

  1. Line 275. The paper by Shimizu et al, was in 2019 or 2009 (ref.66)?

Author Response

We agree that there was not a cohesive link between the animal models of CVD and gut microbiota in CVD. For that reason we have made major revisions to the manuscript. We have removed the animal models section and focused entirely on the gut microbiota. In this revised manuscript rather than focusing on compositional analysis only we have included profiling of microbial metabolic potential and functional genetic alterations in CVD through metagenomic and metabolomics analysis. We discuss potential therapeutic strategies targeting the gut microbiota including probiotics and FMT. We conclude with future perspectives. We hope you will find this revised version much approved.

Reviewer 2 Report

Dear Authors,

After the review process, I have several comments: you should include new data about microbiota fingerprint in CVD (section 4); you should include findings of comparative fingerprinting of the human microbiota in chronic diseases (diabetes) and cardiovascular disease, for example, to observe specific microbial and metabolomic patterns; for this type of paper, you should include a representative figure (or table) in section 4 that increase paper interest; you should include future perspectives of the paper based on the complementary functional strategy for modulation of human gut microbiota.

Best regards.

Author Response

Thank you for your valuable feedback and suggestions which we have taken onboard and feel have much improved the manuscript. We wish to advise also that taking on board feedback from Reviewer 1 we have removed the animal model section and focused entirely on gut microbiota and CVD

  • You should include new data about microbiota fingerprint in CVD (section 4); you should include findings of comparative fingerprinting of the human microbiota in chronic diseases (diabetes) and cardiovascular disease, for example, to observe specific microbial and metabolomic patterns;

This was an excellent suggestion and so in the revised manuscript rather than focusing on compositional analysis only we have included profiling of microbial metabolic potential and functional genetic alterations in CVD through metagenomic and metabolomics analysis.

  • For this type of paper, you should include a representative figure (or table) in section 4 that increase paper interest

We agree and have added a table summarising the main findings of 16S and shotgun sequencing studies examining the gut microbiota and CVD

  • You should include future perspectives of the paper based on the complementary functional strategy for modulation of human gut microbiota.

We discuss potential therapeutic strategies targeting the gut microbiota including probiotics and also FMT. In addition we now conclude the manuscript  with future perspectives.

We hope you will find this revised version much improved.

Round 2

Reviewer 1 Report

This was a review article regarding cardiovascular animal models and gut microbiota, but changed to gut microbiota and cardiovascular disease, which was a big change. The original version was not fluent to read, but this revised version is much better. This reviewer has some comment as described below.

Major comment:

  1. This review article has some sections, such as introduction, gut microbiota in human health. Section number were not in order. The authors should correct them.
  2. Also, the page number was not in order.
  3. The authors put a table of comparison of published studies of gut microbiome in CVD, but also they should put graphical abstract of this review article for better understanding.

Author Response

  1. This review article has some sections, such as introduction, gut microbiota in human health. Section number were not in order. The authors should correct them.The sections have been correctly numbered in the revised manuscript
  2. Also, the page number was not in order.The page numbers have been corrected in revised manuscript
  3. The authors put a table of comparison of published studies of gut microbiome in CVD, but also they should put graphical abstract of this review article for better understanding.This is a great suggestion and we have included a graphic abstract  to aid in a visual summary of the main findings of the article in the revised manuscript. 

Reviewer 2 Report

Dear Authors,

You should make important modifications to the paper structure. Thus, obesity is strictly related to CVD.  In this case, you should include comments based on the last findings published about the link between obesity, microbiota dysbiosis, and neurodegenerative pathogenesis and find a connection with dysbiosis of the CVD group. These data should include more about SCFAs as biomarkers, and the options to combat oxidative stress and inflammatory progression. Best regards.

Author Response

Thank you for your valuable feedback, we have taken all your suggestions on board and feel the manuscript is now greatly improved.

With regards obesity we have added sections on gut microbiota composition, functional analysis, probiotics and FMT related to obesity. 

We have removed the section "Gut microbiota in Health" and instead have incorporated this into our introduction.

We have added a section following the introduction called "Gut Microbiota and CVD" where we discuss in greater detail gut microbiota related metabolites including SCFAs. Here we also discuss oxidative stress and inflammatory progression. We have also included oxidative stress and inflammation in the probiotics section.

In addition we have added a graphic abstract as a visual summary of the main findings of the manuscript. 

Round 3

Reviewer 1 Report

Major comment:

  1. Although the authors replied that they put graphical abstract in this review article, but this reviewer did not find.

Author Response

Dear reviewer.

The GA now is attached.

Round 4

Reviewer 1 Report

This reviewer has no further comment.